

# Effect of acid-suppressive drugs on endoscopic transmural drainage of peripancreatic fluid collections—a randomized controlled trial

Yaoting Li, Tingting Yu, Senlin Hou, Wei Zhang, Haiming Du, Yankun Hou, Jiao Tian and Lichao Zhang

Biliary Pancreatic Endoscopic Surgery, The Second Hospital of Hebei Medical University, Shijiazhuang, Hebei, China

## ABSTRACT

**Background**. Acid-suppressing drugs affect intestinal microbes by inhibiting the secretion of stomach acid. However, it is not clear whether acid suppressive drugs affect the microorganisms in the peripancreatic effusion and affect the postoperative complications.

**Methods**. This study was a prospective randomized controlled trial. A total of 45 patients were enrolled in the trial, and all patients were divided into acid-inhibited and non-inhibited groups. The primary endpoint of our study was to observe the difference in microbiota between the two groups of cyst fluid.

**Results**. A total of 25 patients were included in the acid-inhibited group, and 20 patients were included in the non-acid-inhibited group. There were no significant differences between the two groups in terms of postoperative infection, bleeding, and recurrence rates ($p > 0.05$). In terms of postoperative C-reactive protein levels, the acid-suppressing group was significantly higher than the non-acid-suppressing group ($p < 0.05$). In the 16S microbial sequencing analysis, we found no significant difference in $\alpha$ diversity between the acid-suppressing group and the non-acid-suppressing group, but there was a statistically significant difference in $\beta$ diversity between the two groups.

**Conclusion**. Acid-suppressing drugs can change the microbial composition of pancreatic effusions and increase markers of postoperative inflammation. Acid-suppressive drugs may increase the risk of infection after endoscopic drainage.

## INTRODUCTION

Peripancreatic fluid collections (PFCs) often occur as a result of acute or chronic pancreatitis or pancreatic injury (*Cahen et al., 2005*; *Săftoiu et al., 2013*). Efficient drainage is usually the primary treatment for this condition (*Guo et al., 2016*; *Samuelson & Shah, 2012*). The main treatment modalities include surgical intervention, percutaneous drainage, and endoscopic drainage (*Varadarajulu et al., 2013*; *Saul et al., 2016*; *Tan et al., 2021*; *Xiao et al., 2021*). Multiple studies have demonstrated that endoscopic drainage offers numerous

Corresponding author
Lichao Zhang,
zhanglichao@hebmu.edu.cn

advantages in terms of efficacy, safety, and patient acceptance, and has become a first-line treatment option (*Saul et al., 2016*; *Xiao et al., 2021*; *Varadarajulu et al., 2008*).

Acid suppressive drugs, especially proton pump inhibitors, are widely used in patients with acute pancreatitis worldwide (*Pezzilli et al., 2007*; *Murata et al., 2015*). Proton pump inhibitors inhibit gastric acid secretion by inhibiting H+-K+-ATPase, thereby indirectly inhibiting pancreatic secretion (*Bilski et al., 1992*), and can prevent the occurrence of stress ulcer (*Barkun et al., 2013*). Meanwhile, it was pointed out in a recent study that discontinuation of proton pump inhibitor use reduces the number of endoscopic procedures required for resolution of walled-off pancreatic necrosis (*Powers et al., 2019*). Therefore, many physicians administer acid-suppressing drugs during endoscopic transmural drainage of peripancreatic fluid to reduce the occurrence of bleeding, ulcers, and other complications (*Zhang et al., 2021*). Acid suppressant drugs can change the microbial composition of digestive tract by inhibiting the secretion of gastric acid (*Ma et al., 2020*). At the same time, previous studies have suggested that duodenal microbes are highly similar to pancreatic microbes (*Del Castillo et al., 2019*). However, it remains unclear whether this has an impact on the microbiota of pancreatic fluid, thereby affecting clinical infections. Currently, there is no consensus on the use of acid-suppressing drugs during the perioperative period of endoscopic drainage.

The aim of this study is to investigate the effects of preoperative acid-suppressing drug use in endoscopic transmural drainage of peripancreatic fluid through a randomized controlled trial. Specifically, we aim to assess the impact on the microbiota of pancreatic fluid and postoperative infections, providing a theoretical basis for the clinical application of acid-suppressing drugs.

## MATERIALS AND METHODS

### Study design

This study was designed as a randomized, placebo-controlled, double-blind exploratory trial aiming to investigate the role of acid-inhibiting drug in patients with non-infected peripancreatic fluid collections undergoing endoscopic ultrasound (EUS)-guided transmural drainage of peripancreatic fluid collections from a microbial perspective. This study was designed and conducted in the biliary pancreatic endoscopic surgery of The Second Hospital of Hebei Medical University, from 1st August 2022 to 1st August 2023. The protocol was approved by the Chinese Ethics Committee for Registering Clinical Trials (ChiCTR2100050303) and all research was performed in accordance with the relevant guidelines. This study adheres to the CONSORT guidelines and all authors had access to the study data and reviewed and approved the final manuscript. Following ethical approval from the Second Hospital of Hebei Medical University review board (Ethics Committee of the Second Hospital of Hebei Medical University, Approval No. 2021-R158), consecutive recruitment of patients with peripancreatic fluid collections was conducted in the department of biliary pancreas endoscopy surgery. Written informed consent was obtained from all patients prior to enrollment (minors were subject to informed consent by their guardians). All patients had complete access to all data collected in the study. Portions of this text were previously published as part of a preprint (*Li et al., 2024*).

## Selection criteria

Inclusion criteria:

    1. Patients with a CT or MRI diagnosis of peripancreatic fluid formation.

    2. Patients with indications for endoscopic drainage, including pain, nausea, and jaundice, but not due to infection.

    3. Fluid accumulation persisting for more than 6 weeks and unresponsive to conservative treatment.

    Exclusion criteria:

    1. Patients with spontaneous infection or spontaneous hemorrhage within the cyst.

    2. Patients with contraindications such as portal hypertension and gastrointestinal bleeding.

    3. Patients who have received antibiotics within the past month.

    4. Patients receiving acid-suppressing drugs due to peptic ulcer or other reasons.

    5. Patients with diabetes or autoimmune related diseases and a long history of drug use.

## Study groups

After meeting the inclusion and exclusion criteria, patients in the study group were randomly allocated to either the acid inhibition group (group1) or the non-acid inhibition group (group2). The statistician provided computer-generated randomization assignments using a block randomization method. The random assignments were placed in opaque envelopes and impartially assigned in a 1:1 ratio between the two groups by the ward residents, who were the only individuals with access to the blinded data.

In the acid-suppressive group, patients received acid-suppressive medication (omeprazole 80 mg intravenously) 7 days before endoscopic drainage. In the non-acid inhibition group, patients received a placebo (intravenous saline administration) starting 7 days before the endoscopic transmural drainage procedure. The omeprazole (Hunan Kelun Pharmaceutical Co., LTD., China, 40 mg) used in the acid-suppressing group was 80 mg per day. This is defined by referring to the manufacturer's instructions of the drug (Omeprazole Sodium for Injection, Hunan Kelun Pharmaceutical Co., LTD., China, revised on April 5, 2023) to achieve the best acid-suppressing effect. If a patient presented with symptoms suggestive of infection prior to surgery, antibiotics were administered to ensure patient safety, leading to exclusion from the study. Eventually, a total of 45 patients were enrolled in the study, including 25 patients in the acid inhibition group and 20 patients in the non-acid inhibition group. The visual table is shown in Fig. 1. Patients in the acid-suppressing group stopped using acid-suppressing drugs after drainage. Acid-suppressing drugs were only continued when patients had bleeding, reflux, *etc.*, and required the use of acid-suppressing drugs.

## Procedure and sample collection

All procedures were performed by the same experienced endoscopist. Intravenous anesthesia was administered to all patients. The entire operation process adhered to strict aseptic techniques, with all experiments and surgical materials undergoing rigorous sterilization procedures. Endoscopic ultrasound (OLYMPUS, JAPAN) was utilized for

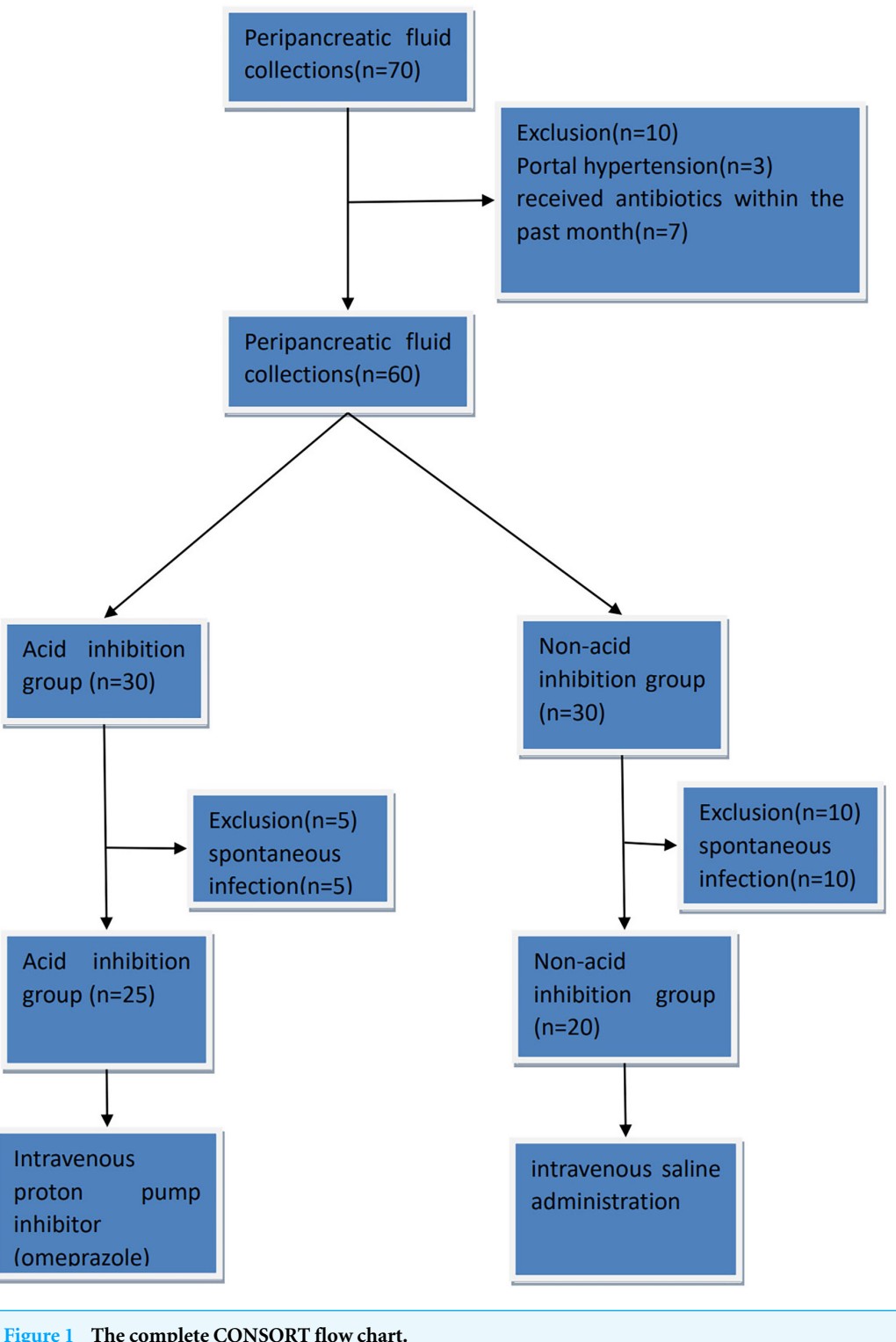

**Figure 1** The complete CONSORT flow chart.

Li et al. (2025), *PeerJ*, DOI 10.7717/peerj.19872  4/19

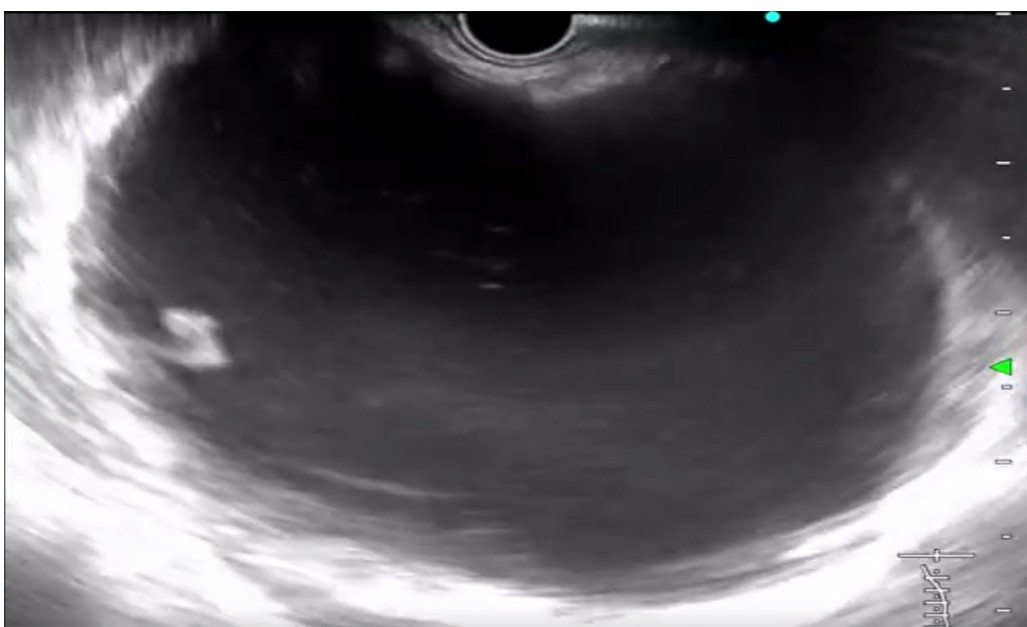

**Figure 2** Endoscopic ultrasonography was used to accurately scan peripancreatic effusion.

precise pancreatic scanning (Fig. 2), ensuring exclusion of bleeding and solid nodules. Subsequently, the cyst was punctured using an ECHO-19 puncture needle (COOK) at a suitable location (either the stomach or duodenum), avoiding blood vessels. Once confirmation of needle entry into the cyst was achieved, 5–10 ml of cyst fluid was extracted using a sterile negative pressure syringe. The collected fluid was then placed in a sterile test tube and promptly stored in a −80 °C refrigerator for subsequent microbiological analysis. Following the puncture, a guide wire was inserted to facilitate cystotomy and channel dilation. Depending on the patient's specific cyst condition, a plastic or metal stent was inserted during the procedure (the procedure was determined by an experienced digestive endoscopist, and this did not affect the study results).

## Definition

Technical success was defined as the successful placement of the stent into the cyst cavity under the guidance of an endoscope and radiography, with no displacement occuring. Clinical success was defined as complete resolution of the effusion or a maximum effusion diameter < two cm, along with the absence of any clinical symptoms. Recurrence referred to the reappearance of fluid accumulation around the pancreas or clinical symptoms occurring 6 months after successful endoscopic drainage. Reoccurrence of effusion within six months is clinical treatment failure and is not defined as recurrence. Bleeding was determined by the presence of hematemesis, hematochezia, or a decline in hemoglobin of >10 g/L within 72 h post-surgery. As long as there is a body temperature greater than 38.5 degrees celsius after surgery, regardless of whether there is an increase in white blood cells, it is defined as infection. Once infection is confirmed, treatment is given immediately

with intravenous antibiotics (ceftriaxone 2g, intravenous). The primary outcome measure focused on differences in microbial species and abundance between the two groups, with clinical infection, white blood cell count, and C-reactive protein were secondary outcomes.

## Metabarcoding of cyst fluid

Soil DNA kit (Omega) was used to extract genomic DNA from peripancreatic effusion samples. The DNA was measured by 0.8% agarose gel electrophoresis and quantified by ultraviolet spectrophotometer. The DNA is stored in a $-20\,°C$ refrigerator for subsequent analysis after passing the test. In order to ensure the sequencing quality, the optimal insertion segment for sequencing was 200–450 bp according to the Miseq sequence read length. In this study, V3~V4 regions of 16S rRNA gene were selected for sequencing. The polymerase chain reaction (PCR) primers were 338F (5′-ACTCCTACGGGAGGCAGCA-3′) and 806R (5′-GGACTACHVGGGTWTCTAAT-3′). Conventional PCR thermal cycle procedures: initial denaturation at 98 °C for 30 s, denaturation at 98 °C for 15s, annealing at 50 °C for 30 s, extension at 72 °C for 30 s. The above three processes were carried out for 25 cycles, finally extended for 5 min at 72 °C, and finally stored conventionally at 4–10 °C. Amplification results were performed on 2% agarose gel electrophoresis. After the detection was qualified, the target fragment was removed, and the target fragment was recovered with the oxygen gel recovery kit. The PCR products were quantified using the Quant-iTPico Green dsDNA Assay Kit on a Microplate reader (BioTek, FLx800) and then mixed according to the amount of data required for each sample. Illumina truseq was used to construct sequencing library of the PCR products obtained. Qualified sequencing library (index sequence cannot be repeated) was diluted according to gradient, then mixed according to sequencing amount, denatured into single strand with NaOH, and sequenced on the machine. The barcode V3–V4 amplicon was sequenced by Illumina Miseq. 200–450 bp was selected as the best sequencing length. Preliminary screening was conducted according to overlapping bases, and the pairing of the peer sequences was performed using FLASH software [14]: the overlapping base lengths of Read 1 and Read 2 sequences were required to be $\geq 10$ bp, and base mismatch was not allowed. Then the valid sequence is obtained according to the Index information corresponding to each sample. Finally, QIIME software (Quantitative Insights Into Microbial Ecology, v1.8.0) was used to identify error sequences and remove them. The amplification and sequencing of 16S rRNA gene was completed by Personal Biotechnology Co., LTD. (Shanghai).

## Statistical analysis

The main outcome of our study was to observe the difference of microbiota in the two groups of cyst fluid. According to previous studies (*Weinroth et al., 2022*; *Kelly et al., 2015*), there is no unified scheme for sample size determination and efficacy determination of microbial sequencing studies. Therefore, the calculation of our sample size is based on the non-metric multidimensional scaling analysis (NMDS) and beta diversity analysis required for this study, as well as previous similar studies. When there are > four samples per group, it can be analyzed and produce meaningful results. In order to improve the credibility of the study, we defined the sample size as 45 cases. This sample size can satisfy

the reliability of the study. Statistical analyses were performed using IBM SPSS 27.0 and R 3.4.4 with a test level of $\alpha = 0.05$. Due to our small sample size, we used standardized mean differences (SMDs) to assess inter-group differences. A value less than 0.3 is considered to have little difference between groups. Alpha diversity was analyzed using QIIME2 (2019.4), R language, ggplot2 package, and Kruskal-Wallis rank sum test and dunn 'test as post hoc tests to verify the significance of differences. Shannon index, Simpson index, Chao1 index and observed species index were used to analyze the $\alpha$ diversity between the two groups. The beta diversity was analyzed using scikit-bio package, R language, vegan package implementation. The Adonis method was used to analyze the differences in beta diversity between groups. The explainability (R2) and significance (P) of the grouping scheme to the variance of the distance matrix were calculated through the vegan package of R, and the number of permutation tests was set to 999.

## RESULT

### Patient characteristics

The study conducted from 1st August 2022 to 1st August 2023 screened 70 patients with peripancreatic effusion, out of which 45 eligible patients were enrolled. Among them, there were eight men and 37 women, with a mean age of 49.17 years (range 17–75). The main reasons for exclusion were portal hypertension, recent use of acid-suppressing drugs or antibiotics and spontaneous infection.

### Comparison of clinical data between the acid inhibition group and non-acid inhibition group

Table 1 presents the general clinical data of the acid inhibition and non-acid inhibition group. The acid inhibition group consisted of 25 patients, while the non-acid inhibition group consisted of 20 patients. There were no significant differences in gender, age, cyst location, cyst type, stent type, puncture location and cyst diameter between the two groups (SMD < 0.3). These results indicated that there were no significant differences in preoperative clinical characteristics between the two groups and were comparable.

### Clinical results
#### Short-term clinical outcome
The technical success rate of both groups was 100%, and no adverse events occurred during the operation. There were two clinical failures in both the acid-inhibited and non-inhibited groups, with similar clinical success rates (92% *vs* 90%, $p = 1$). All patients who failed endoscopic treatment were successfully treated by surgery and endoscopic intervention again.

Postoperative bleeding occurred in five patients in the acid-inhibited group and three patients in the non-acid-inhibited group, with no statistically significant difference (20% *vs* 15%, $p = 0.965$). All bleeding patients were successfully treated with conservative and endoscopic hemostasis, and no adverse events such as death occurred. In terms of infection, no significant difference was found in clinical infection rates between the two groups (52% *vs* 45%, $p = 0.641$). There was no significant difference between the two groups in the

**Table 1   Comparison of preoperative general conditions between the two groups.**

| Variable | Acid inhibition group ($n = 25$) | Non-acid inhibition group ($n = 20$) | SMD |
|---|---|---|---|
| Age (years) | $57.36 \pm 14.18$ | $50.25 \pm 15.31$ | $-0.25556$ |
| Sex (male) | 18/25 (72%) | 13/20 (65%) | 0.04308 |
| Cyst diameter (cm) | $11.09 \pm 2.35$ | $15.57 \pm 4.96$ | $-0.04416$ |
| Cyst type (PPC) | 21/25 (84%) | 15/20 (75%) | $-0.02701$ |
| Cyst location (body and tail) | 23/25 (92%) | 18/20 (90%) | 0.068318 |
| Location of puncture (stomach) | 24/25 (96%) | 20/20 (100%) | 0.04714 |
| Type of stent (metal stent) | 2/25 (8%) | 4/20 (20%) | $-0.06832$ |
| Etiology | Alcohol 5/25 (20%) Trauma 2/25 (8%) Gallstones 18/25 (72%) | Alcohol 4/20 (20%) Trauma 1/20 (5%) Gallstones 15/20 (75%) | $-0.0364$ |

**Table 2   Postoperative adverse events were compared between the two groups.**

| | Acid inhibition group ($n = 25$) | Non-acid inhibition group ($n = 20$) | P |
|---|---|---|---|
| Technical success | 100% | 100% | 1 |
| Clinical success | 23/25 (92%) | 18/20 (90%) | 1 |
| Infection | 13/25 (52%) | 9/20 (45%) | 0.641 |
| Bleeding | 5/25 (20%) | 3/20 (15%) | 0.965 |
| Recurrence | 7/25 (28%) | 1/20 (5%) | 0.107 |

**Table 3   Comparison of postoperative infection indexes between the two groups.**

| | Acid inhibition group ($n = 25$) | Non-acid inhibition group ($n = 20$) | P |
|---|---|---|---|
| White blood cell count ($10^9$/L) | 11.6 (6.3–19.5) | 10.4 (6.5–14.6) | 0.189 |
| C-reactive protein (mg/L) | 148.44 (89–270) | 109.5 (54–178) | 0.008 |

white blood cell count ($P = 0.189$). However, C-reactive protein was significantly different between the two groups after operation ($P = 0.008$).

### Long-term clinical outcome

The median follow-up time was 12 months, and recurrence occurred in seven patients in the acid-inhibited group and only one patient in the non-inhibited group, with no statistical difference ($P = 0.107$). This is statistically insignificant likely due to the sample size but there is a trend in data. As we know that acid reduction occurs in pancreatic walled-off pancreatic necrosis (WOPN). All patients with recurrence were successfully treated by endoscopic re-intervention. Detailed data are presented in Tables 2 and 3.

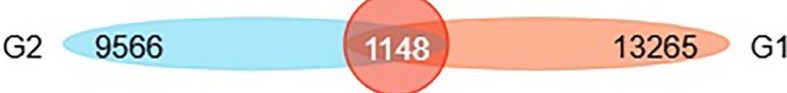

G1:Acid inhibition group
G2:Non-acid inhibition group

**Figure 3** **The Venn diagram reveals the similarity and overlap of two groups of OTU numbers.**

## Comparison of microbial characteristics between acid inhibition group and non-acid inhibition group
### OTU level analysis
This study performed operational taxonomic unit (OTU) clustering on non-repeated sequences (excluding single sequences) according to 97% similarity, and chimeras were removed during clustering to obtain representative sequences of OTUs. The total number of OTUs in the two groups was 22,831. The number of overlapping OTUs between the two groups is 1,148. The similarity and overlap of the OTU number and composition of each group can be visualised in a Venn diagram (Fig. 3).

### Species composition analysis
The proportion of annotations for each taxonomic level of OTUs was counted to obtain the relative abundance of each species at each taxonomic level. The proportion of annotations for each taxonomic level of OTUs was counted to obtain the relative abundance of each species at each taxonomic level. 16S ribosomal RNA (rRNA) data from the metagenome sequences (Fig. 4A) showed that at the phylum taxonomic level, Firmicutes, Proteobacteria, Bacteroidetes and Actinobacteria were dominant in both the acid-inhibited group and non-acid-inhibited group. The proportion of Firmicutes in the two groups was 36.41% and 33.03%. The proportion of Proteobacteria in the two groups is 27.69% and 30.24%. Respectively, bacteroidetes accounted for 12.11% and 10.43% in the two groups. The proportion of Actinobacteria in the two groups is 10.88% and 10.44%. The taxonomic hierarchy tree more intuitively expresses the species composition of the two groups (Fig. 4B).

### Alpha diversity analysis
The diversity indices (Chao 1, Simpson and Shannon) were applied to reflect its alpha diversity. The Shannon, Simpson, and Chao 1 demonstrated that there was no striking difference in the microbial community abundance between groups ($p > 0.05$) (Fig. 5). This showed that there was no significant difference in microbial abundance in the sac fluid between the acid-inhibited group and the non-acid-inhibited group.

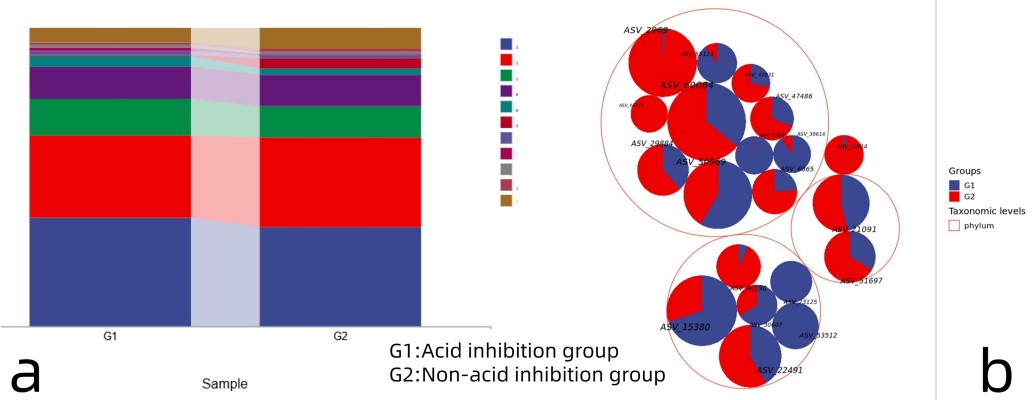

G1:Acid inhibition group
G2:Non-acid inhibition group

**Figure 4** (A) Taxonomic composition analysis of the two groups of microorganisms. (B) Hierarchical tree of microorganism classification in two groups.

### Beta diversity analysis

As can be seen in Fig. 6A, the acid-inhibited group and non-acid-inhibited group can be clearly distinguished. Meanwhile, the results of the Adonis analysis based on weighted UniFrac showed statistically significant differences in the microbial community structure between the two groups ($R2 = 0.05$, $p = 0.021 < 0.05$) (Fig. 6B).

Principal coordinate analysis of all samples (weighted UniFrac index) revealed that the difference in microbial composition of the cyst fluid, and the microbial composition of the two groups were heterogeneous (Fig. 7).

### Microbial difference analysis

We analyzed the bacterial differences between the two groups using the random forest method. Spirochaetes, Firmicutes and Bacteroidetes were significantly enriched in the acid-inhibited group compared with the non-acid-inhibited group. In the random forest map (Fig. 8), spirochaetes showed the greatest difference between the two groups.

## DISCUSSION

Our study included a total of 45 patients, and the two groups were well-matched in terms of demographic characteristics, etiology of pancreatitis, cyst types, and cyst locations (Table 1). Both groups achieved similar rates of clinical success and technical success ($p > 0.05$), which are consistent with previous studies (*Siddiqui et al., 2013*; *Sousa et al., 2021*). Similarly, there were no significant differences between the two groups in terms of postoperative infection, bleeding, and recurrence rates ($p > 0.05$). Regarding the postoperative infection indicator, there was no difference in white blood cell count between the two groups. However, in terms of postoperative C-reactive protein levels, the acid-suppressing group was significantly higher than the non-acid-suppressing group ($p < 0.05$). We conducted 16S microbial sequencing analysis to investigate the microbial composition in the cyst fluid, and we found no significant difference in α diversity between the acid-inhibited

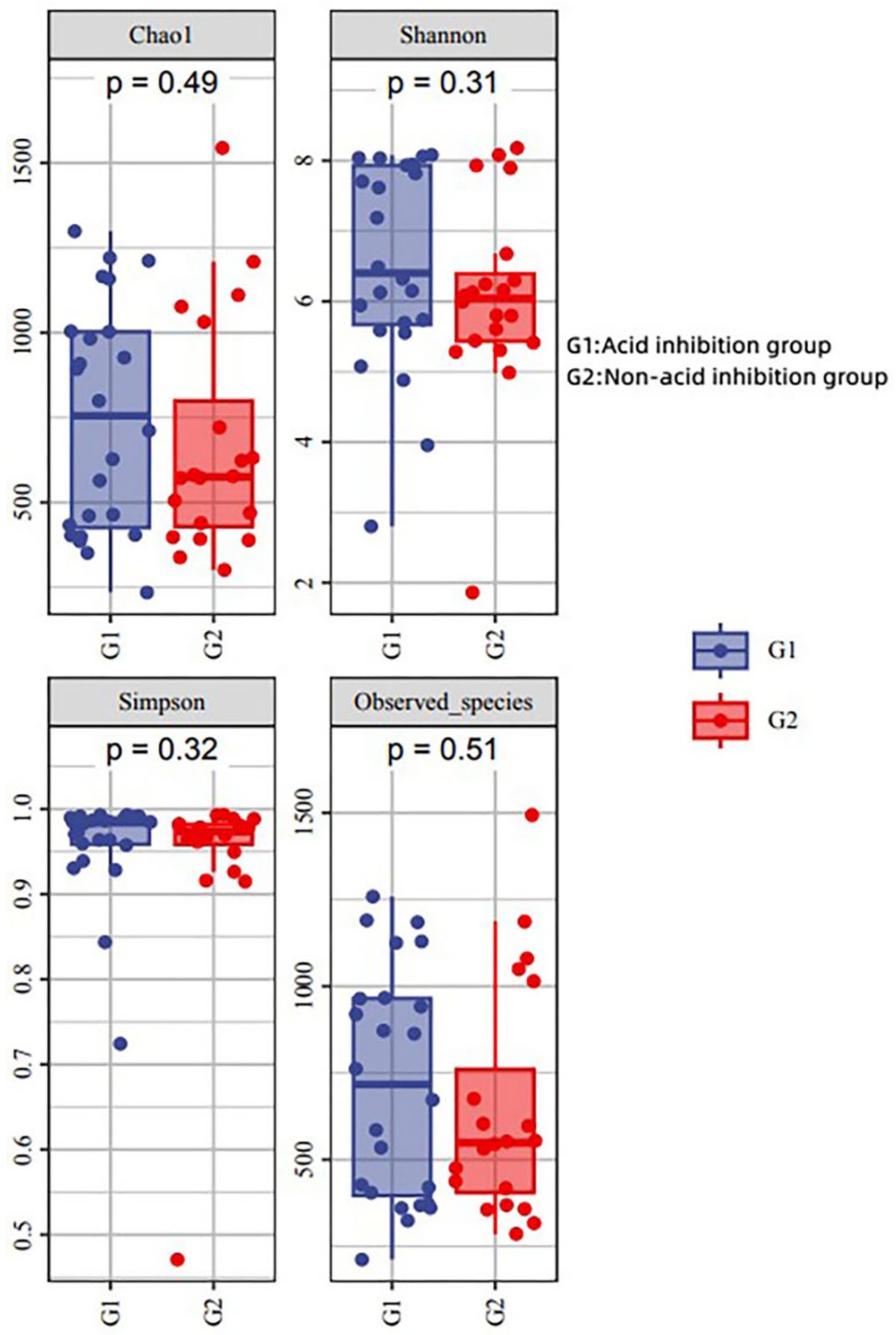

**Figure 5** Analysis of alpha diversity in two groups.

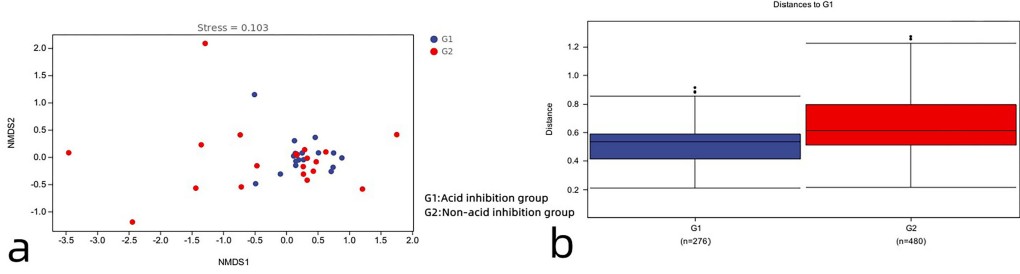

**Figure 6** (A) The NMDS diagram reveals the differences between the two groups. (B) Analysis of β diversity difference between acid-inhibited group and non-acid-inhibited group.

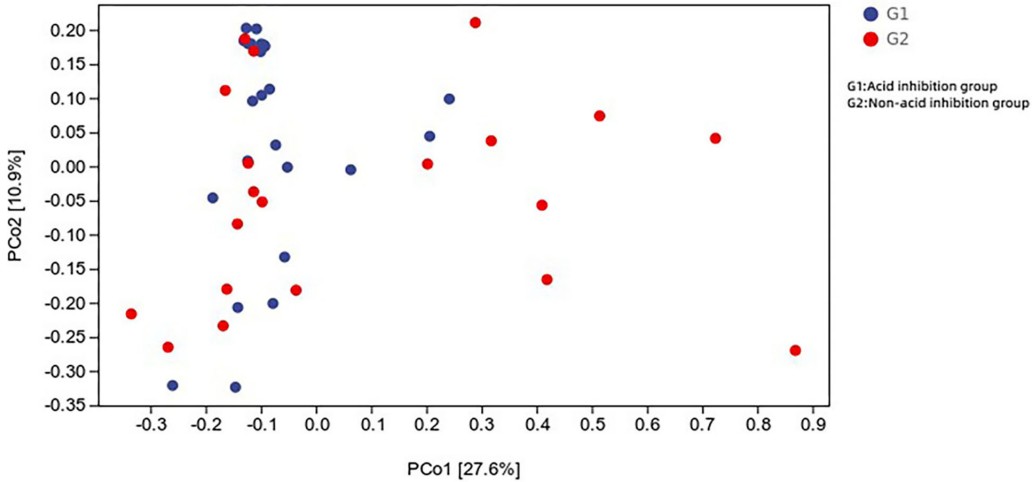

**Figure 7** Distance matrix and principal coordinates analysis (PCoA) analysis further revealed the difference between the two groups.

group and the non-acid-inhibited group. However, there was a significant difference in β diversity analysis between the two groups.

Postoperative infection following endoscopic transmural drainage of peripancreatic fluid may have multiple causes. Some studies have indicated that the etiology of pancreatitis and the size of the cysts are important factors contributing to postoperative infections (*Guo et al., 2016*). Larger cysts are more challenging to drain effectively and are more prone to infection (*Guo et al., 2016*). Some guidelines recommend the prophylactic use of antibiotics (*Khashab et al., 2015*). However, a recent randomized controlled trial determined that antibiotics were unnecessary for aseptic cyst drainage (*Jagielski et al., 2022*). Both studies essentially conclude that maintaining adequate drainage is crucial for infection prevention. Nevertheless, even when the drainage remains unobstructed, some patients may still experience symptoms of infection, such as fever. This suggests that microbes play a significant role in this process. Our study confirmed that acid-suppressing drugs can alter the microbial composition of peripancreatic effusions and increase markers of

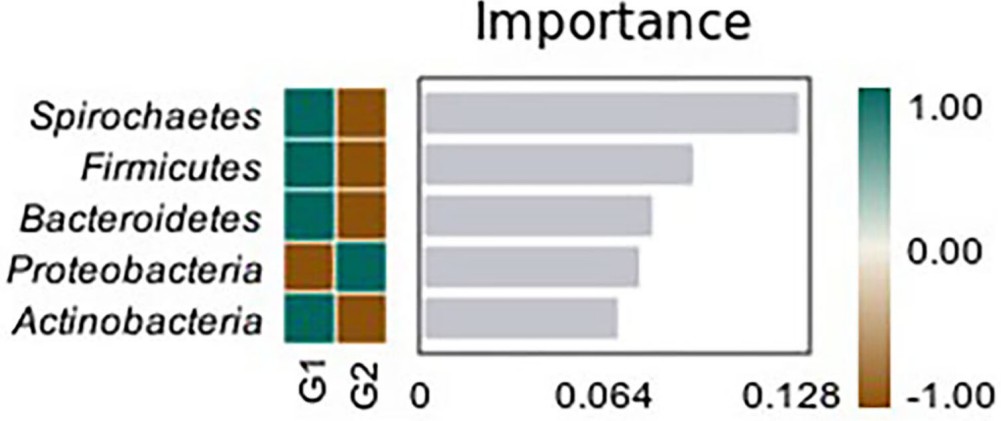

**Figure 8** **Random forest maps reveal differences in microbiota between the two groups.**

postoperative infection, such as C-reactive protein. During the development stage of acute pancreatitis inflammation, intestinal bacterial displacement is common due to the damage of the intestinal mucosal barrier and microbial imbalance (*Lankisch, Apte & Banks, 2015*; *Zhou et al., 2019*; *Zhang et al., 2023*; *Wu et al., 2023*). However, studies have shown that normal pancreatic tissue is not sterile and possesses similar microbiological characteristics to the duodenum (*Del Castillo et al., 2019*; *Pushalkar et al., 2018*; *Sethi et al., 2018*). This implies a microbial interaction between the normal duodenum and pancreas, though the mechanism remains unclear. In an animal trial, fluorescence was observed in pancreatic tissue after mice were administered fluorescently labeled Enterococcus faecalis, suggesting bacterial migration from the gut to the pancreas (*Pushalkar et al., 2018*). Additionally, some studies indicate that bacteria can infiltrate the pancreas through the pancreatic duct (*Shirai & Tsukada, 2024*). Therefore, we believe that this microbial interaction persists even after the formation of fluid encapsulation in the later stages of pancreatitis. Acid suppressive drugs can inhibit gastric acid secretion and alter the microbial composition of the duodenum, thereby affecting the microbial composition of the cyst fluid through the interaction between the pancreas and duodenum.

Acid-inhibiting drugs are widely used in clinic. But the problem of microbial imbalance is also gradually emerging. Previous studies have shown that acid-suppressing drugs can change the composition of microbes in the gut (*Jackson et al., 2016*), and more oral bacteria have been found in the stool of patients taking acid-suppressing drugs (*Imhann et al., 2016*). At the same time, studies have found that excessive use of acid-inhibiting drugs increases the risk of colonizing carbapenem-resistant bacteria in the gut (*Lee et al., 2024*). In a meta-study, the overgrowth of small intestine bacteria caused by acid-suppressing drugs

should also be taken into account (*Lo & Chan, 2013*). The microflora in peripancreatic effusion is mostly caused by intestinal microbial migration during the development of pancreatitis. Whether the use of acid suppressants causes changes in the microbiota in the peripancreatic effusion is unknown.

As shown in the random forest map, Spirochaetes, Firmicutes and Bacteroidetes were significantly different between the two groups, and Spirochaetes, Firmicutes and Bacteroidetes were significantly enriched in the acid-inhibiting group. Previous research has suggested that an imbalance between Firmicutes and Bacteroidetes may exacerbate inflammatory responses (*Wu et al., 2023*; *Pushalkar et al., 2018*). This increased inflammatory response may be achieved through the metabolites of the flora. At present, studies have shown that the decrease of bacteria producing butyric acid can aggravate the progression of acute necrotizing pancreatitis, and butyric acid may play an anti-inflammatory role through the STAT 1/AP1-NLRP 3 pathway by inhibiting HDAC1 (*Van den Berg et al., 2021*). In addition, butyrate can reduce pancreatic damage during acute pancreatitis (AP) by eliminating inflammatory factors and inhibiting NLRP-3 inflammatory vesicles (*Pan et al., 2019*). Lactic acid, a metabolite of bifidobacterium, can alleviate the progression of acute pancreatitis by regulating TLR 4/MyD 88 and NLRP 3/Caspase 1 pathways (*Li et al., 2022*). Acetate produced by Parabacteroides can alleviate the severity of acute pancreatitis by reducing neutrophil infiltration (*Lei et al., 2021*). Similar conclusions were obtained in our study, and the postoperative C-reactive protein level was significantly elevated in the acid-suppressed group compared to the non-acid-suppressed group. Although there was no effect on postoperative clinical infections, this may be due to the limited sample size. The changes of microorganisms in the capsule fluid did not cause significant clinical problems, but the imbalance of microecology caused by acid-suppressing drugs should be paid attention to. This may be related to an underlying clinical infection.

As an exploratory study, this investigation has two primary limitations. Firstly, the specific bacterial species responsible for infection susceptibility were not identified. Secondly, since the primary focus of the study was on microbial differences, the sample size was relatively small. More large-scale studies are needed in the future to determine the role of microbiota in infection.

## CONCLUSION

Acid-suppressing drugs can change the microbial composition of pancreatic effusions and increase markers of postoperative inflammation. Acid-suppressive drugs may increase the potential infection after endoscopic drainage.

### Funding

The research was supported by the Natural Science Foundation of Hebei Province (H2021206439). The funders had no role in study design, data collection and analysis, decision to publish, or preparation of the manuscript.

### Grant Disclosures

The following grant information was disclosed by the authors:
Natural Science Foundation of Hebei Province: H2021206439.

### Competing Interests

The authors declare there are no competing interests.

### Author Contributions

- Yaoting Li conceived and designed the experiments, performed the experiments, analyzed the data, prepared figures and/or tables, authored or reviewed drafts of the article, and approved the final draft.
- Tingting Yu conceived and designed the experiments, performed the experiments, analyzed the data, prepared figures and/or tables, and approved the final draft.
- Senlin Hou conceived and designed the experiments, performed the experiments, analyzed the data, prepared figures and/or tables, and approved the final draft.
- Wei Zhang conceived and designed the experiments, performed the experiments, prepared figures and/or tables, and approved the final draft.
- Haiming Du conceived and designed the experiments, performed the experiments, prepared figures and/or tables, and approved the final draft.
- Yankun Hou conceived and designed the experiments, performed the experiments, authored or reviewed drafts of the article, and approved the final draft.
- Jiao Tian conceived and designed the experiments, performed the experiments, authored or reviewed drafts of the article, and approved the final draft.
- Lichao Zhang conceived and designed the experiments, performed the experiments, authored or reviewed drafts of the article, and approved the final draft.

### Human Ethics

The following information was supplied relating to ethical approvals (*i.e.*, approving body and any reference numbers):

The Ethics Committee of the Second Hospital of Hebei Medical University approved the study (2021-R158).

### Data Availability

Sequence data is available at the National Genomics Data Center: CRA020366 https://bigd.big.ac.cn/gsa/browse/CRA020366.

Raw data is available at Zenodo:

Zhang, L., & The Second Hospital of Hebei Medical University. (2025). Effect of acid suppressive drugs on peripancreatic effusion microorganism [Data set]. Zenodo. https://doi.org/10.5281/zenodo.15617810.

### Clinical Trial Registration

The following information was supplied regarding Clinical Trial registration:

ChiCTR2100050303.

## Supplemental Information

Supplemental information for this article can be found online at http://dx.doi.org/10.7717/peerj.19872#supplemental-information.

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
