# Peer review of "Effect of acid-suppressive drugs on endoscopic transmural drainage of peripancreatic fluid collections—a randomized controlled trial"

_PeerJ, doi:10.7717/peerj.19872_

## Round 0.1 · original submission · Major Revisions

·

Basic reporting

This is a novel idea which explores the effect of acid suppression before PFCs drainage. All previous reports discussed the role of halting acid suppressive medications after transluminal drainage.
Also it discusses this from the microbial point of view.
The article is well structured and the study is well designed to lead to scientifically sound conclusion.
There are few flaws which can be corrected before publishing.
Page 6 there was mention of antibiotic and non antibiotic groups which should be corrected to acid suppressive medications or non acid suppressed groups.
Page 7 please provide dedicated reference for lines 55 n and 56.
Please clarify if the patient is going to continue on acid suppression after drainage.
Please clarify the refrence of the specific dose of Omeprazole used in acid suppressed group.

Why did you analyse the alpha and beta diversity between both groups?

Regarding table 1 could you please add the mean and SD for each group.
Table 2 please rectify the 2nd item.
Thanks with my best wishes.

Experimental design

.

Validity of the findings

.

·

Basic reporting

This is a nice clinical trial investigating the rate of infection and resolution of peripancreatic fluid collections post pancreatitis. It is a double blinded randomized controlled clinical trial. Statistical analysis was thoroughly and nicely performed accounting for multiple factors. The findings can be significant and of great benefit for future practices.

We had prior data, and the most prominent one was by Powers et al (PMCID: PMC6589997) which showed that discontinuation of PPI in Walled off pancreatic necrosis can lead to faster resolution and less necresctomy procedures. Which I recommend citing this article as well.

Comments:
1) I would subclassify the cyst types as pancreatic pseudocyst is different in approach and management from walled off pancreatic necrosis as well as in component of microbes. If the authors meant one subtype, then this should be mentioned clearly in the manuscript.

2) There is a discrepancy in timing of the study. Abstract says August 2022-2023 and then Methods in line 211 mentions June 2022-2023?

3) In line 214: I thought recent antibiotic use was an exclusion and not acid reducer use.

4) In line 239: which group had higher CRP? Would include that here. Also CRP is non specific acute phase reactant and marker of inflammation and since patients had pancreatitis, this maybe the reason of elevated CRP. I would have suggested to obtain CRP on presentation and day, 3 days, 7 days after the endoscopic drainage.

5) In line 242: would recommend rewriting this as there is a trend here in the result 7 compared to 1 even if not statistically significant clinically can be significant depending on the cyst type and need of subsequent interventions. I would recommend adding that this is statistically insignificant likely due to the sample size but there is a trend in data. As we know that Acid reduction in pancreatic WOPN.

6) In Line 303: I believe there was no mention of redividing the groups into antibiotic group vs no antibiotic group. Was there further stratification? Or was it meant to say Acid reduction vs. no? This was also mentioned in the abstract. Also I believe that recent antibiotic use was already an exclusion criteria.

Experimental design

- Design was good but would work on further characterize the arms of the study. Also mention if there was further stratification.

Validity of the findings

I do think the sample size is small and needs larger scale studies to get significant results but it is Novel approach and may lead to change practices on larger scales.

Additional comments

None

---

## Round 0.2 · accepted · Accept

The authors have addressed all of the reviewers' comments and manuscript is ready for publication.

·

Basic reporting

After the authors addressed the proposed revisions, the manuscript looks good for publication.

Experimental design

After the authors addressed the proposed revisions, the manuscript looks good for publication. Experimental design is good and meets standards.

Validity of the findings

After the authors addressed the proposed revisions, the manuscript looks good for publication. The findings are novel and worth publishing.